# Polariton Lasing in Micropillars With One Micrometer Diameter and Position-Dependent Spectroscopy of Polaritonic Molecules

U. Czopak[1*], M. Prilmüller[1], C. Schneider[3], S. Höfling[2], G. Weihs[1]

**1** Institut für Experimentalphysik, Universität Innsbruck, Innsbruck, Austria
**2** Technische Physik and Wilhelm-Conrad-Röntgen Research Center for Complex Material Systems, Universität Würzburg, Würzburg, Germany
**3** Institute of Physics, University of Oldenburg, 26129 Oldenburg, Germany
* ulrich.czopak@uibk.ac.at

February 9, 2021

## Abstract

**Microcavity polaritons are bosonic light-matter particles that can emit coherent radiation without electronic population inversion via bosonic scattering. This phenomenon, known as polariton lasing, strongly depends on the polaritons' confinement. Shrinking the polaritons' mode volume increases the interactions mediated by their excitonic part, and thereby the density-dependent blueshift of the polariton to a higher energy is enhanced. Previously, polariton lasing has been demonstrated in micropillars with diameters larger than three microns, in grating based cavities, fiber cavities and photonic crystal cavities. Here we show polariton lasing in a micropillar with one micron diameter operating in a single transverse mode that can be optimally coupled to a singlemode fiber. We geometrically decouple the excitation with an angle from the collection. From the number of collected photons we calculate the number of polaritons and observe a blueshift large enough to qualify our device for novel schemes of quantum light generation such as the unconventional photon blockade. To that end, we also apply angled excitation to polaritonic molecules and show site-selective excitation and collection of modes with various symmetries.**

## 1 Introduction

Strong optical nonlinearities are a key component for many quantum information processing applications [1]. Conventional lasers are nowadays widely used for linear optical communication, however, the intrinsic Poissonian photon number distribution of a laser limits its usefulness for most quantum protocols that aim to process individual quanta of light [2]. By confining light in tiny resonators that are comparable in size to its wavelength, the light-matter interaction in the solid can be strongly enhanced. For example, quantum dots embedded in the antinode of a micropillar cavity brought great advances towards perfect single photon sources [3], but scaling up to multiple indistinguishable devices is difficult because the spatial and spectral alignment of the quantum dots is non trivial. Excitons in two-dimensional

quantum wells and atomically thin crystals can couple strongly to a cavity's radiation field resulting in hybrid light-matter polariton modes [4–6]. In polaritonic micropillars many nonlinear many-body physics phenomena have been demonstrated, including polariton lasing [7,8], and just recently, first indications of single quantum effects have been observed in highly engineered fiber cavities [9,10].

Microcavity exciton-polaritons result from a superposition of a cavity photon and a quantum well exciton, i.e. a bound electron-hole pair [11, 12]. These bosonic quasi-particles effectively introduce an optical nonlinearity to the radiation field inherited from the electronic interactions of their matter part. In contrast to conventional lasers that need to be driven to an electronic population inversion, polariton lasers emit coherent light through enhanced bosonic scattering into a common state [7,13,14]. Given the light mass polaritons inherit from their photonic part, this effect, which is closely related to Bose-Einstein condensation [15], already happens at moderately cryogenic temperatures in gallium arsenide, and even at room temperature for organic polaritons and atomically thin crystals with their much tighter bound excitons [16–19]. The threshold for the onset of enhanced coherence is much lower without the need for a population inversion, and the scattering into a common state enables measuring the particle-density dependent blueshift.

Confining polaritons to a small volume enhances the electronic interactions, alters the light-matter coupling and thereby lowers the lasing threshold and increases the blueshift [20,21]. The ultimate goal would be a blueshift per polariton that is bigger than its linewidth, paving the way for achieving the so-called photon blockade [22] in the solid state [23], which filters single photons out of a conventional laser's radiation. For the blockade to become strong, it was calculated that the polariton mode volume must not be larger than a cubic wavelength in the material [23,24]. An alternative approach termed the unconventional photon blockade uses the interference of two weakly nonlinear coupled modes to achieve photon number squeezing [25]. Strong photon antibunching is predicted even if the blueshift is two to three orders of magnitude smaller than the linewidth [26]. However, as the required mean input power versus the resulting antibunching strength scales with the nonlinearity, increasing the nonlinearity is still desirable. We show here that the nonlinear blueshift is indeed much enhanced by confining polaritons in small micropillars. In addition the expected weak signal can be optimally coupled to single mode fibers and fast and sensitive single photon detectors, as micropillars have an excellent mode overlap with singlemode fibers [3].

In this article we study polaritonic micropillars with diameters ranging from 3 µm down to 1 µm and analyze their nonlinear optical properties. To do so, we employ excitation under an angle to the sample plane. This technique was originally developed to excite quantum dot nanowires [27]. We show here that this in principle filter-free excitation technique also works for micropillar cavities with diameters down to 1 µm, making our approach attractive for experiments with quantum dots in micropillars as well. The angled excitation allows us to directly address the quantum wells of any pillar on the sample while collecting all the light they emit perpendicular to the sample plane. This approach makes it easier to optically address tiny structures close to the optical diffraction limit, which is otherwise challenging in cryogenic environments. We observe polariton lasing accompanied by a blueshift of 28 µeV per polariton in a 1 µm-sized pillar with a minimal linewidth of 149(5) µeV. In addition, we also apply the angled excitation to study so-called polaritonic molecules. We show that we can selectively excite symmetric and antisymmetric polaritonic modes that occur in such overlapping micropillars [28]. In similar systems fascinating nonlinear many-body physics has been shown, including polariton condensation [29], Josephson oscillations and

self-trapping [30], bistability [31] and periodic squeezing [32]. Studying polaritonic molecules with geometrically decoupled excitation and collection demonstrates a promising avenue to investigate single-particle nonlinearities mediated by the unconventional polariton blockade.

The structure of the paper is as follows: After an introduction to microcavity polaritons we describe our experimental setup and semiconductor sample. Subsequently, we present our results for lasing in single pillars, and our results for the position-dependent excitation of polaritonic molecules.

## 2   Theory

Polaritons are bosonic mixed light-matter particles resulting from the strong coupling between a quantum well exciton (creation operator $b^\dagger$) at energy $E_X$ and a microcavity photon (creation operator $a^\dagger$) with energy $E_C$. The effective Hamiltonian reads [23]:

$$H_{\text{eff}} = E_X b^\dagger b + E_C a^\dagger a + \hbar\Omega_R(ba^\dagger + b^\dagger a) + \frac{V_{XX}}{2}b^\dagger b^\dagger bb - V_S(b^\dagger b^\dagger ab + a^\dagger b^\dagger bb) \tag{1}$$

with $\Omega_R$ being the vacuum Rabi frequency, i. e. the rate at which exciton and photon are strongly coupled and exchange energy. This rate depends on the exciton's dipole moment, which is a function of the design of the quantum wells, its' material and dimensions, and the electric field strength which depends on the cavity's dimensions. A measurement on an unetched part of our cavity yields a value of $\Omega_R \approx 6\,\text{meV}$. However, it should be noted that the Rabi-splitting is expected to be altered by tighter photonic confinement due to light-induced changes of the exciton radius [33]. The smaller the confining photonic structure is, the lower is the number of cavity eigenmodes. If the coupling rate is stronger than the individual decay rates, strong coupling forms a polaritonic mode for each photonic mode. The polariton inherits the nonlinear interaction $V_{XX}$ from its exciton part, mainly due to the exchange of carriers [34, 35], and an additional, weaker nonlinearity is caused by the saturation of the exciton's oscillator strength $V_S$ [36]. For a two-dimensional exciton with binding energy $E_B$ and Bohr radius $a_B$ the interaction constants read

$$\frac{V_{XX}}{A} \approx 6E_B a_B^2; \frac{V_S}{A} = \frac{8\pi}{7}\hbar\Omega_R a_0^2 \tag{2}$$

for a system area $A$ [33]. The binding energy $E_B$ and the Bohr radius $a_B$ are in general a function of the quantum well's material and dimensions [37]. In (In)GaAs quantum wells the Bohr radius is about $5\,\text{nm}$ to $15\,\text{nm}$ and the binding energy between $4\,\text{meV}$ to $30\,\text{meV}$, depending on the well's thickness and material composition. If we take the values from [37] for a $3\,\text{nm}$ thick quantum well we have $E_B = 30\,\text{meV}$ and $a_B = 5\,\text{nm}$ which yields $V_{XX} = 4.5\,\mu\text{eV}\,\mu\text{m}^2$ and $V_S = 2.7\,\mu\text{eV}\,\mu\text{m}^2$. The resulting polariton-polariton interaction for the here considered lower polariton in one spin configuration depending on the exciton fraction $X$ (0.8) and photon fraction $C$ (0.2) is [12] $V_{LP} = |X|^4 V_{XX} + 2|X|^2 XC V_S = 2.4\,\mu\text{eV}\,\mu\text{m}^2$. In Reference [23] a factor of $2.67/(2R)^2$ was suggested for a cylindrically confining potential, with $R$ being the radius of the cylinder. This yields blueshifts of $6.4\,\mu\text{eV}$, $1.5\,\mu\text{eV}$ and $0.7\,\mu\text{eV}$ per polariton for micropillars with diameters of $1\,\mu\text{m}$, $2\,\mu\text{m}$, and $3\,\mu\text{m}$, respectively. In addition, an incoherent blueshift from excitons created by the pump that do not couple to the light field can be significant [38]. Details about the dynamics behind polariton lasing can be found in [12, 13, 39].

# 3 Setup for angled excitation

In order to realize decoupled excitation and collection, the Janis ST-500 flow cryostat is mounted on a horizontal translation stage and so allows moving the sample with respect to the fixed aspheric collection lens with a numerical aperture of 0.68 mounted above (see figure 1). In this way, all the radiation emitted from the pillar can be collected.

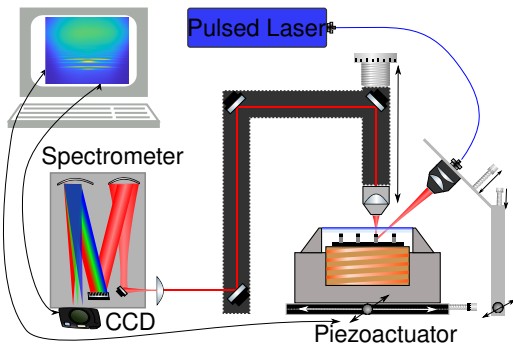

Figure 1: The setup allows moving the micropillar sample in both transverse directions, coarsely by fine adjustment screws and by computer-controlled piezo actuators for fine adjustments. The excitation can be aligned independently by a single mode fiber with a focuser at the output mounted on translation stages next to the cryostat. A fixed asphere collimates the radiation from the micropillars and sends it to a spectrometer where the emission is recorded. For the position dependent spectroscopy in section 4.3 an automated measurement sets the piezo position and takes a spectrum for each position.

A long-distance micro-focuser delivers the beam for the angled excitation (Schäfter und Kirchhoff 5M) from an attached polarization-maintaining optical fiber. A Titanium::Sapphire laser with a repetition rate of $80\,\mathrm{MHz}$ at a wavelength of $820\,\mathrm{nm}$ ($1.512\,\mathrm{eV}$) and a pulse length of $10\,\mathrm{ps}$ is used for above-band excitation. Because the focal length of the focusing lens $f_{\mathrm{focus}}$ is smaller than that of the collimating lens $f_{\mathrm{coll}}$ the resulting spot size is smaller than the mode field diameter MFD of the singlemode fiber ($\Phi_{\mathrm{Spot}} = \frac{f_{\mathrm{focus}}}{f_{\mathrm{coll}}}\mathrm{MFD}$). In our case this corresponds to $\approx 2\,\mu\mathrm{m}$, if we do not consider distortions due to the cryostat window. The whole excitation optics is mounted on a five-axis stage designed in a way to control the excitation in all three translational dimensions plus two angles (see figure 1). A spectrometer (Acton SP 2750) is used to measure the spectrum of the light emitted by the nanostructures on a nitrogen-cooled CCD camera. This allows us to measure the number of photons resolved in spectral photon energy and consequently to estimate the nonlinear interaction strength of the polaritons and the resulting blueshift.

In order to estimate the number of polaritons that contribute to one measured spectrum we calculate the spectrometer efficiency to be 9.9(36)%. A summary of all factors is given in table 1 below.

| Component | Efficiency |
|---|---|
| 3x Al+MgF$_2$ Mirror | 0.87(3) |
| 1500 Grooves/mm Grating | 0.62(5) |
| Aperture Loss / Mode Match | 0.75(25) |
| CCD Quantum Efficiency | 0.65(5) |
| Electronic Gain | 0.50(5) |
| Total | 0.099(36) |

Table 1: Factors contributing to the spectrometer efficiency

Thus $\approx 10(4)$ photons in front of the spectrometer cause one CCD count during its minimal exposure time of $12.5\,\mathrm{ms}$. Due to our decoupled excitation and collection, the collecting asphere directly collimates the emission towards the spectrometer. In between we only use dielectric mirrors and anti reflection coated windows and lenses, so the loss is negligible. The Aperture Loss corrects for the entrance slit of the spectrometer and the filling factor of the spherical reflectors in the spectrometer, which depends on the pillars emission angle.

## 3.1 Sample

Our cavities consist of two GaAs/AlGaAs DBR mirrors grown by molecular beam epitaxy (MBE). A $\lambda$-cavity between the two DBR mirrors hosts six $3.3\,\mathrm{nm}$ wide InGaAs quantum wells with $10\,\%$ indium content, separated by $10\,\mathrm{nm}$ thick GaAs barriers. This rather exotic design with closely neighboring, very thin quantum wells leads to strong light-matter coupling resulting in a Rabi frequency of $\Omega_R \approx 6\,\mathrm{meV}$. Micropillar structures are created by reactive ion etching using a mask defined by electron beam lithography. The micropillars range in diameter from $1\,\mu\mathrm{m}$ to $5\,\mu\mathrm{m}$, and for each diameter so-called photonic molecules consisting of two overlapping pillars with center-to-center distances ranging from $50\,\%$ to $100\,\%$ of the pillar diameter are defined. Because the MBE growth rate decreases from the center of the wafer to its edge, we achieve a wide range of cavity lengths and thus different exciton-photon detunings.

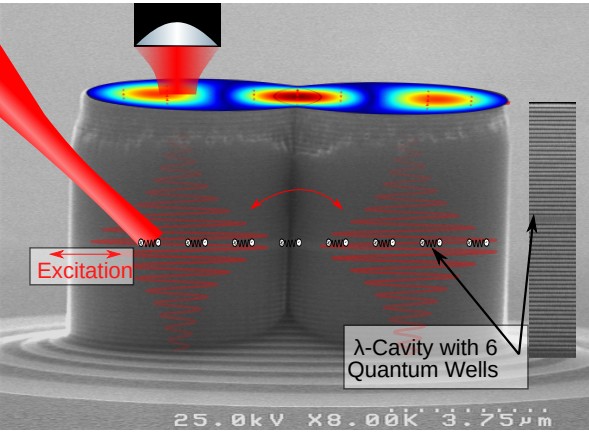

Figure 2: An SEM picture of a polaritonic molecule with a pillar diameter of 3 µm and a center-to-center distance of 2.4 µm (see scale bar). The $\lambda/4$ layers of the mirrors as well as the cavity spacer are clearly visible in the inset on the right. The field intensity profile of the second symmetric mode is superimposed on the top surface to illustrate the influence of the position-dependent excitation and collection. Excitons in the quantum wells as well as individual pillar modes and tunneling between them are sketched.

## 4  Results

### 4.1  Polariton lasing

To investigate the nonlinear effects in single polaritonic micropillars we start our investigation by scanning the power of the excitation laser (described in Section 3) and analyze the recorded spectra. We rotate a half-wave plate in front of a polarizer to control the laser power and record spectra for a series of excitation powers. The maximum laser power measured directly after the excitation lens is approximately 4 mW. The actual power injected into the quantum wells, however, is a function of the excitation spot size relative to the pillar diameter and the size of the quantum wells and is thus significantly smaller. The coupling is further reduced by wavefront abberations caused by the convergent laser beam passing through the cryostat's window and the small effective cross section of the quantum wells when projected onto the laser beam axis. In Figure 3 a), c), and e) we show a representative selection of spectra together with a Gaussian fit to the polariton mode with the lowest energy. Although the cavity line should have a Lorentzian shape, the observed peaks show better overlap with Gaussian functions. In the low power spectra the onset of lasing is visible, while in the high power ones the shape of the peaks with maximum blueshift can be seen. From the Gaussian fits we extract the peak emission intensity (normalization constant) and the linewidth (FWHM) and plot these against a linearized excitation power axis in figure 3 b), d), and f). The blueshift is analyzed in the next chapter (Section 4.2).

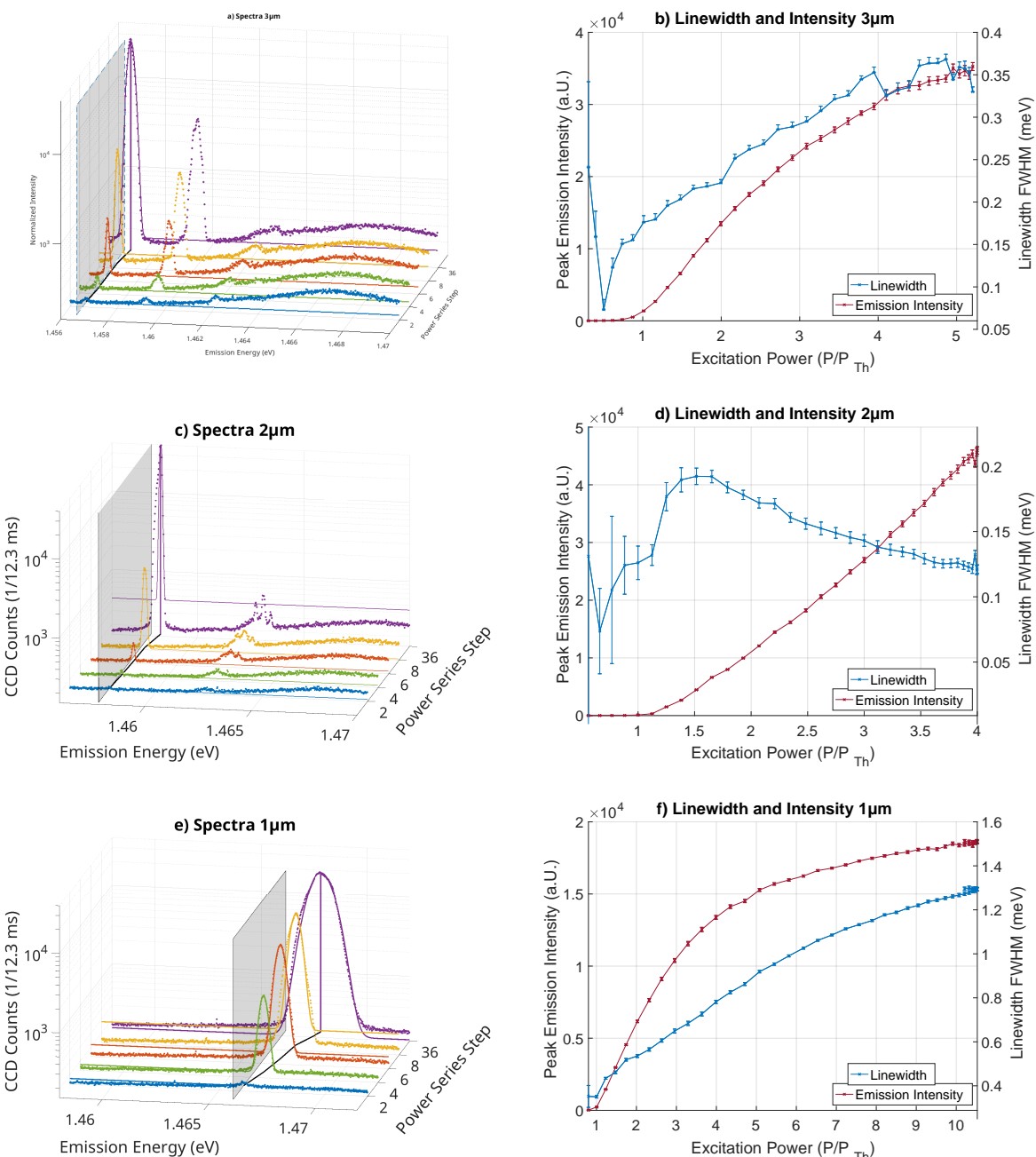

Figure 3: In a), c), and e) we show spectra recorded for different excitation powers from micropillars ranging from $3\,\mu\mathrm{m}$ to $1\,\mu\mathrm{m}$ in diameter. The grey layer indicates the center of the emission at the lowest excitation power to serve as a guide to the eye for the relative blueshift at higher powers. While for the $3\,\mu\mathrm{m}$ pillar there are still three lower polariton modes present, single mode operation becomes evident in the $1\,\mu\mathrm{m}$ pillar. Due to the photonic confinement the lowest energy mode is at higher energy for smaller pillars. At about $1.468\,\mathrm{eV}$ a broad peak from excitons that do not couple to the cavity is visible. For the lowest power this is the dominant emission feature, then higher order polariton modes get populated and at highest excitation powers most of the population is in the fundamental mode. The linewidth and peak height of the Gaussian fits to the fundamental mode are shown in b), d), and f). Above a certain threshold excitation power $P_{\mathrm{Th}}$ the peak height grows in a nonlinear way accompanied by a decline in linewidth. Both are evidence for polariton lasing. The linewidth broadening at higher powers is caused by temporal integration over the ringdown from higher to lower blueshifts as the population decays after the excitation pulse.

The effect of shrinking pillar sizes on the nonlinear emission properties can be clearly seen in Figure 3. For smaller pillars fewer modes are present and the fundamental photonic mode gets blueshifted. This effect makes the polaritons more exciton-like, which in addition to the higher exciton-exciton interaction energy in a smaller reservoir enhances the blueshift. The high and low power emission peaks overlap at the low energy side of the spectrum because we integrate over the ringdown from high to low occupation. Therefore, one would expect an asymmetric peak shape that shows the actual linewidth on its high energy side. This can be seen to some extent for the high power spectrum of the 2 µm pillar (purple line in Figure 3 c). This effect also seemingly broadens the linewidth for higher excitation powers and to extract more meaningful values of the linewidth we fit Gaussian functions to the high energy side of the peaks. For the two bigger pillars we see a decline in linewidth while approaching the lasing threshold, which is characteristic for polariton lasing, and a broadening thereafter caused by the blueshift and the ringdown to lower energies. In the 1 µm pillar the lasing threshold is so low that the linewidth already starts at a minimal value. The lowest value for the linewidth we measure here is $73\,\mu\text{eV}$ (FWHM) corresponding to a $Q$-factor of 20000(1570). In the 2 µm pillar however the linewidth seems to decrease again for powers over 1.5 times the threshold. This could be related to some mode competition as the fundamental mode here is much more enhanced at higher powers than in the 3 µm pillar, while no higher modes exist at all in the 1 µm pillar. In addition, in the fit to the highest power spectrum the asymmetric peak shape is even less compatible with a Gaussian than for the other data. The singlemode nature of the 1 µm pillar lowers its lasing threshold in comparison to its bigger counterparts, and also increases the interaction-related blueshift. Due to this extreme blueshift the peak broadens more in width than it grows in height. This also explains the bigger linewidth that we observe in this structure, as the particle number fluctuation is expected to obey Poissonian statistics [40]. The fact that we see lasing in such a tiny structure demonstrates the high quality of our semiconductor sample and optical setup. To the best of our knowledge our device is among the smallest polariton lasers reported in the literature so far [21, 41, 42]. The extreme blueshift that we observe motivates an estimate of the number of contributing polaritons, which we perform in the following section.

## 4.2   Polariton Interactions: Blueshift

In order to determine the absolute magnitude of the blueshift it is crucial to know how many polaritons contributed to it. To this end we follow references [20, 43] and count the photons that are recorded by the spectrometer. As outlined in Section 3 we can roughly estimate the number of photons in front of the spectrometer if we multiply the total counts that contributed to a peak by 10(4). We calculate the total counts by summing over all the CCD photoelectron counts in a peak corrected for the background level. To obtain the number of photons per pulse we have to divide the total counts by the number of laser pulses during one CCD exposure time $12.3\,\text{ms} \cdot 80\,\text{MHz} = 984000$. Assuming equal photon and exciton contributions we expect half the polaritons to decay non-radiatively as excitons and the other half to exit the cavity as photons and enter the spectrometer. Plotting the blueshift from the fits of the power series against the number of polaritons we infer an estimate of the nonlinearity per particle. A possible caveat here is our off-resonant excitation that creates an excitonic background, which also blueshifts the polaritons. Another possible problem are potential dark background states, which could in principle live much longer than the $12.3\,\text{ns}$ between two laser pulses and cause an additional blueshift [44].

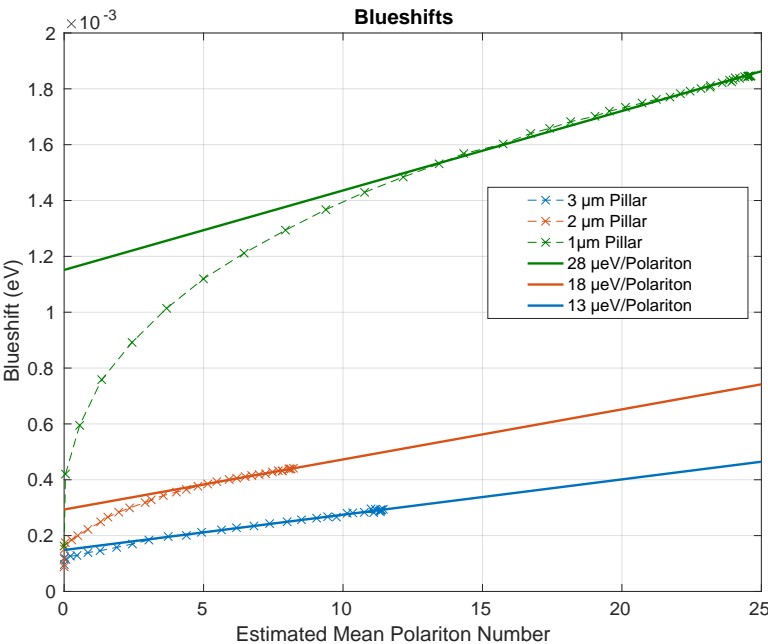

Figure 4: Plotting the center energy shift as a function of the extimated occupation number shows that smaller pillars exhibit a much bigger blueshift than larger ones. The solid lines are linear fits to a subset of data points at high occupation number to extract the asymptotic behavior.

The blueshifts in Figure 4 seem to show a logarithmic dependence on the occupation number, as it was also seen in Reference [8]. However, for very low powers the majority of the excitations in the quantum well are bare excitons, as can be seen in the low power spectra in Figure 3. The absolute amount of these uncoupled excitons is difficult to estimate because they are not expected to decay into the cavity mode that is coupled into the spectrometer. From the slope of the linear fits we get a blueshift of $28\,\mu eV$, $18\,\mu eV$ and $13\,\mu eV$ per polariton, for the $1\,\mu m$, $2\,\mu m$ and $3\,\mu m$ pillar respectively. Our measurements agree well with the theoretical predictions for interaction constants of exciton-polaritons and are within the broad range of results reported in the literature so far. The inverse proportionality between the nonlinearity and the transverse confinement, as discussed in the theory section (2), is clearly demonstrated. A recent overview of comparable measurements can be found in [43]. In comparison to that reference our results are an order of magnitude higher, but significantly higher nonlinearities have been reportet for InGaAs quantum wells. We emphasize that we use a structure with a quite exotic arrangement of InGaAs quantum wells. They are much thinner and closer to each other than what is commonly used, and the changes of the exciton's properties and light coupling are difficult to estimate theoretically, especially given the tight spatial confinement in our micropillars. Polariton lasing was mostly studied using GaAs quantum wells, although InGaAs offers certain advantages for polariton lasing [45]. Therefore, observing such a strong blueshift here is surprising and encouraging.

In previous works it was assumed that the threshold for lasing occurs at a mean occupation of one. This matches our findings, as we find a occupation of around one between the third

and the fourth measurement point for the 1 μm pillar and between the seventh and the eigth for the 2 μm and 3 μm pillars, exactly at the lasing threshold indicated in figure 3 b, d, e. Although it would be interesting to determine the absolute threshold excitation power this is not possible here due to limitations discussed in section 4.1. The results obtained motivate further, more sophisticated measurements. Suggested improvements include to resonantly excite polaritons with a repetition cycle lower than the lifetime of any dark states in the system and to precisely calibrate the intensity measurement. Resonant excitation under an angle from the side is a big advantage towards that goal because then all the radiation leaving the pillar can be collected.

In summary in this section we used our angled excitation to show lasing in single pillars with diameters ranging from 1 μm to 3 μm. Features of lasing were shown for each and the dependence of the magnitude of the blueshift on the pillar size was clearly demonstrated. In addition it could be interesting to study the dynamics of the power-dependent emission in the multimode pillars for various exciton-photon detunings and excitation locations, but this is beyond the scope of this paper.

## 4.3   Position-dependent spectroscopy

In addition we perform position-dependent spectroscopy on polaritonic molecules such as the one in Figure 2. This is similar to what was shown in [29], however we continuously change the excitation location, and in addition our approach makes it possible to excite one pillar and collect from the other. The excitation and collection lenses are kept stable and the polaritonic molecule is moved along its main axis by a piezo actuator in the translation stages below the cryostat. Depending on the individual mode's field strength at the excitation location different modes are excited. For example, if the molecule is excited at the center between the two pillars, no antisymmetric modes are excited and detected because they have a node there. We demonstrate this effect by showing two different spectra of a molecule with a pillar diameter of 3 μm and center-to-center separation of 2.4 μm. One spectrum is recorded for excitation at the center between the two pillars and the other at the center of one pillar (Figure 5a). The big advantage of this approach is demonstrated by scanning the cryostat's position along the molecules axis, thereby collecting spectra for each position. The recorded spectra assembled together in a two dimensional density plot clearly show the mode structure of the polaritonic molecule under investigation. As in a simple double well potential, the fundamental mode is symmetric, followed by an anti-symmetric mode with one node in the center, then another symmetric mode with two nodes symmetric around the center and so on. Thus, modes with similar symmetry appear at similar excitation locations (see figure 6).

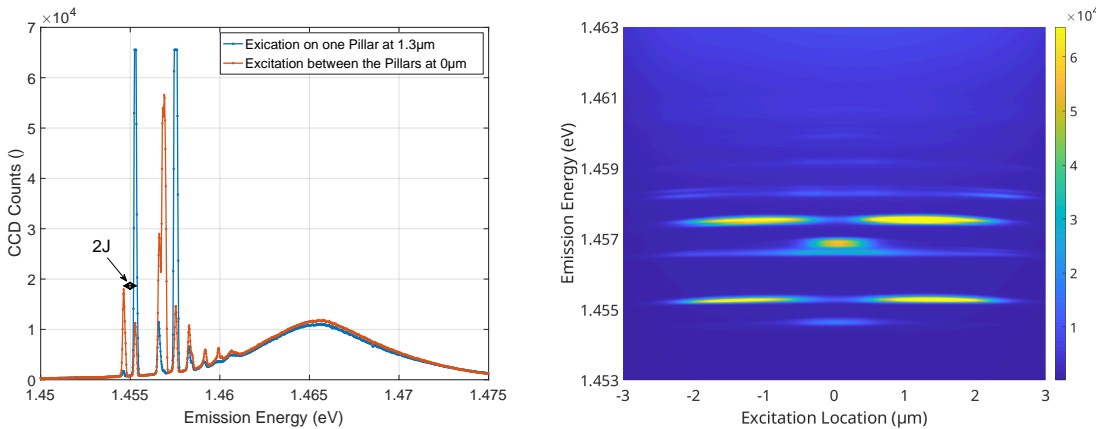

Figure 5: The orange line in the left plot shows a spectrum taken when we excite a polaritonic molecule in its center between the two pillars. The symmetric fundamental mode is stronger in this case. The blue line, in contrast, shows a spectrum recorded when we excite the molecule central on one pillar. In this case antisymmetric modes are stronger. On the right side we assemble 98 spectra in a density plot and interpolate between them. The colorbar shows the respective CCD counts.

In this way we can study the mode structure of polaritonic molecules and measure the tunneling constant. From the data above we exctract a value of J=340 μeV.

To validate our experimental results and understand the physical mechanisms we performed simulations in Lumerical FDTD Solutions. Dipoles that excite a purely photonic molecule with same dimensions as in our experiment are swept along 99 positions on the molecule axis and for each position a spectrum is computed. This corresponds to our measurement where we excite excitons site-selectively along the molecule's axis.

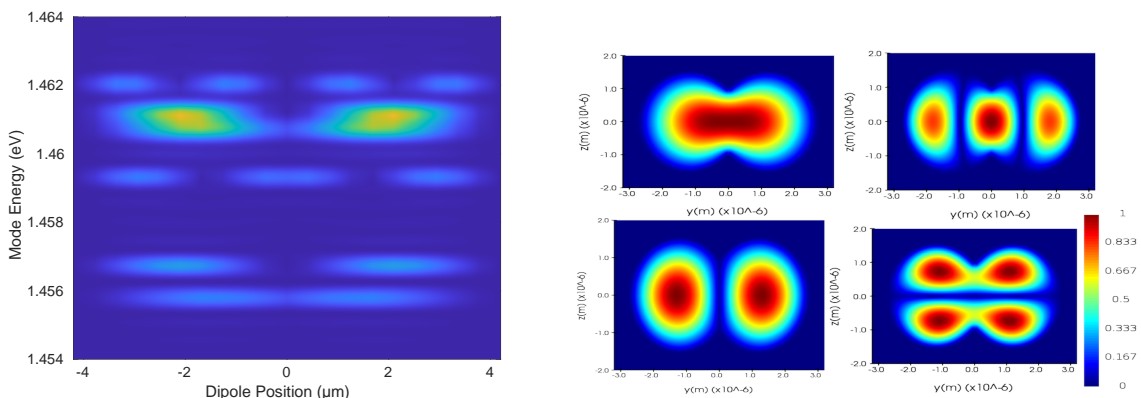

Figure 6: The FDTD Simulation shows qualitative similar results as the measurements when we move the dipoles that excite the modes on the molecules axis and assemble the spectra together as before. For illustration, on the right side we show the field amplitude of lowest order modes computed in the eigenmode solver.

Both individual and the assembled spectra one clearly sees the first two branches of sym-

metric and antisymmetric modes, separated by an energy splitting that depends on the pillar separation and is equal to two times the tunnel constant $J$. An important difference between measurement and simulation is that the dipoles in the simulations are point like emitters, whereas in the measurement we create excitons over the whole spot size. Additionally, in a polaritonic molecule the exciton-photon detuning plays a big role in the dynamics that determine which mode gets populated most strongly. Our technique can be used to routinely scan entire samples, facilitated by independent control over excitation and collection. By further improving the optical performance, for example by positioning the lens inside the cryostat, it should be feasible to selectively drive desired modes at one location on the molecule resonantly and collect from the other one. This could pave the way to more sophisticated experiments such as the unconventional polariton blockade.

# 5   Conclusion

We used angled excitiation of polaritonic micropillars and observed their nonlinear behavior in polariton lasing. In the 1 µm sized pillar we observed an extremely strong blueshift of 28 µeV per polariton, witness to high nonlinearities and a promising result towards the unconventional photon blockade. In addition, we applied our excitation technique to polaritonic molecules and compared the results to simulations. Further improving the optical performance, for example by placing the lenses inside the cryostat, angled excitation could also work for the resonant case. That is also interesting for micropillars hosting quantum dots as there is no need to filter the pump light then. We hope that the presented techniques and experiments open the way to studies with strongly interacting polaritons in the single particle regime.

# Acknowledgements

We acknowledge help in setting up the FDTD Simulations by S. Betzold. Support by K. Winkler, M. Emmerling and A. Wolf in sample fabrication is acknowledged. We thank R. Keil, R. Chapman and S. Frick for helpful comments to the manuscript. N. Gulde form Roper Scientific we thank for informations about the spectrometer.

**Author contributions**   UC performed the measurements, analyzed the data, wrote the manuscript and evaluated the simulation. MP helped to set up the measurement. CS and SH designed and fabricated the micropillar sample. GW supervised the project. All authors contributed comments to the manuscript.

**Funding information**   We thank the Austrian Science Fund FWF for supporting this work through project I2199. UC and MP received funding from the Austrian Science Fund (FWF), Grant No. W1259, "DK-AL"

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
