# Peer review of "Polariton Lasing in Micropillars With One Micrometer Diameter and Position-Dependent Spectroscopy of Polaritonic Molecules"

_SciPost Physics_

## Round 1 · Referee Report · Anonymous · 2021-4-24

Report

The authors report experiments on InGaAs micropillar structures of two types: single micropillars and two coupled micropillars. Experiments are performed under non-resonant excitation using an original angle scheme. One of the main claims of the manuscript is the observation of lasing in a one-micron diameter micropillar, and a measurement of the polariton-polariton interaction constant that differs significantly from previously measured values.
As I will discuss in detail below, the main claims are not supported by the experimental evidence and the analysis of the data presents serious flaws. For these reasons I do not recommend publication of the manuscript.

The main claim and main novelty of the paper is the observation of polariton lasing in a one-micron micropillar. Polariton lasing in such small structures has not been previously reported. The main reason for this lack of observations is most likely the dominant role of nonradiative processes in such small sized etched structures. Figure 3(e) and (f) show the experimental evidence on which the claim is based here. However, no hint of lasing can be seen in these figures. There is no identifiable threshold, there is no increase of temporal coherence (i.e., linewidth narrowing) and there is not any superlinear increase of the emitted power. Moreover, at low power, the linewidth of the 1micron structure shown in Fig. 3 is about 3 to 4 times larger than for the 2 and 3 microns structures. How can it be that with more loses the absolute power threshold can be much smaller, as mentioned by the authors in the main text? An increase of the interaction strength could account for part of it, but such a large increase of losses seems to be the main effect here, which should result in a dramatic increase of the threshold (and not a decrease).
By the way, Fig. 3(f) shows linewidths of about 400microeV at low power, while the main text mentions linewidths of 75microeV for the smallest micropillar.

Another important remark is the discussion on the extraction of the polariton-polariton constant. The discussion is full of contradictions and it cannot be accepted in a rigorous scientific article.
First, there is missing information on the estimation of the number of photons emitted from the system. What is the exciton-photon detuning of the lowest energy polariton modes? Is the structure symmetric, that is, does it emit equal number of photons on the substrate and epitaxial sides? The authors mention “Assuming equal photon and exciton contributions we expect half the polaritons to decay non-radiatively as excitons and the other half to exit the cavity as photons and enter the spectrometer.” What is the meaning of this? What kind of non-radiative process is assumed here? And what the half-half ratio is based on?
Second, the authors admit that “A possible caveat [to extract the polariton interaction constant from the blueshift] here is our off-resonant excitation that creates an excitonic background, which also blueshifts the polaritons”. The presence of this reservoir in non-resonant excitation schemes is dominant up to very high powers. This has been documented in a large number of works (see [1] to cite one). I do not understand how can one admit that this contribution is important and no take it into account in any attempt to estimate the polariton-polariton interaction constant. The dominant presence of this reservoir is probably the reason why the numbers obtained here, 28-13microeV per polariton, are much larger than in previous experiments [2,3]. Another question is how to compute this number given that excitons from different QWs might lightly interact. This is important to compare the extracted interaction constant in the present manuscript with the numbers reported in other works, which are usually defined per QW (in standard microcavity samples, only polaritons with excitons in the same QW can interact).

Regarding the second part of the manuscript devoted to the study of photonic modes in a two-coupled micropillar system, it is not clear what is the novelty with respect to published works. The spectroscopy of the lowest energy modes of polaritonic molecules has been reported before with spatial data of very high quality [4]. In addition, there are significant differences between the experimental profiles of some modes, particularly the lowest energy one in Fig. 5-right, and the FDTD simulation. What is the reason?

One minor point: fits and linewidth analysis of Fig. 3(a), (c) and (e) are hard to follow in the figures.

[1] L. Ferrier, E. Wertz, R. Johne, D. D. Solnyshkov, P. Senellart, I. Sagnes, A. Lemaître, G. Malpuech, and J. Bloch, Interactions in Confined Polariton Condensates, Physical Review Letters 106, 126401 (2011).
[2] A. Delteil, T. Fink, A. Schade, S. Höfling, C. Schneider, and A. İmamoğlu, Towards Polariton Blockade of Confined Exciton–Polaritons, Nature Materials 18, 219 (2019).
[3] G. Muñoz-Matutano, A. Wood, M. Johnsson, X. Vidal, B. Q. Baragiola, A. Reinhard, A. Lemaître, J. Bloch, A. Amo, G. Nogues, B. Besga, M. Richard, and T. Volz, Emergence of Quantum Correlations from Interacting Fibre-Cavity Polaritons, Nature Materials 18, (2019).
[4] M. Galbiati, L. Ferrier, D. D. Solnyshkov, D. Tanese, E. Wertz, A. Amo, M. Abbarchi, P. Senellart, I. Sagnes, A. Lemaitre, E. Galopin, G. Malpuech, and J. Bloch, Polariton Condensation in Photonic Molecules, Phys. Rev. Lett. 108, 126403 (2012).

---

## Editorial Decision

editor-in-charge_assigned